# A Survey on the Evaluation of Monosodium Glutamate (MSG) Taste in Austria

**DOI:** 10.3390/foods14010022

**Published:** 2024-12-25

**Authors:** Emilia Iannilli, Emilise Lucerne Pötz, Thomas Hummel

**Affiliations:** 1Department of Psychology, Karl-Franzens-Universität Graz, 8010 Graz, Austria; 2Smell & Taste Clinic, Department of Otorhinolaryngology, Technische Universität Dresden, 01307 Dresden, Germany

**Keywords:** text sentiment analysis, sensory analysis, cultural acceptance, perception

## Abstract

The umami taste is well validated in Asian culture but remains less recognized and accepted in European cultures despite its presence in natural local products. This study explored the sensory and emotional perceptions of umami in 233 Austrian participants who had lived in Austria for most of their lives. Using blind tasting, participants evaluated monosodium glutamate (MSG) dissolved in water, providing open-ended verbal descriptions, pleasantness ratings, and comparisons to a sodium chloride (NaCl) solution. Discrimination tests excluded MSG ageusia, and basic demographic data were collected. A text semantic-based analysis (TSA) was employed to analyze the emotional valence and descriptive content of participants’ responses. The results showed that MSG was predominantly associated with neutral sentiments across the group, including both female and male subgroups. “Sour” was the most frequent taste descriptor, while “unfamiliar” characterized the perceptual experience. Pleasantness ratings for MSG and NaCl were positively correlated, suggesting that overcoming the unfamiliarity of umami could enhance its acceptance and align it with the pleasantness of salt. These findings advance the understanding of umami sensory perception and its emotional and cultural acceptance in European contexts, offering valuable insights for integrating umami into Western dietary and sensory frameworks.

## 1. Introduction

The umami taste is a relatively new primary taste. It was first described by Ikeda in 1909 [1], and only recently, specific umami taste receptors were discovered by Nelson et al. in 2002 [2].

The evolutional relevance of the umami taste is demonstrated by the high levels of free glutamate contained in breastfeeding milk [3,4], by the fact that human infants are equipped with umami receptors already at birth [4], and by the fact that l-glutamate is a palatable taste for human infants [5].

Umami sensory quality studies have shown that the taste of MSG is significantly enhanced when combined with nucleotides such as inosine-5′-monophosphate (IMP) and guanosine-5′-monophosphate (GMP) [6] (Yamaguchi, 1998), indicating the epigenetic significance of umami in signaling dietary proteins.

From a health perspective, studies have shown that MSG can help in reducing dietary sodium intake, gaining an advantage due to its synergistic effect with saltiness [7,8,9,10,11]; it can enhance satiety in the context of protein intake [12,13,14]; it can sustain fullness for a more extended period than other treatments [15]; it can control infant feeding [16,17,18]; and last but not least, it can play a crucial role in reducing the bitterness typical of green leaves [19,20], which is beneficial for our health but also leads to a lower impact on the environment. The evidence mentioned here is only a tiny part of the plethora of scientific publications proving glutamate’s benefit in dietary intake.

Umami food palatability enhancement has led to the use of l-glutamate-rich foods for centuries in many cultures, including ancient Greece and Rome, with fermented foods or rich umami vegetables being mixed with fish and meats.

Besides being available in natural sources, umami has been made available as monosodium glutamate (MSG), a salt of glutamic acid very similar to cooking salt [1] (Ikeda, 1909). Since then, MSG has been used in food industries, restaurants, and at home to improve the taste and palatability of meals.

Asians, especially the Japanese, are familiar with the taste of MSG and regularly add it to their food. On the contrary, despite the long history of experience in combining the umami taste in natural foods and spices—see, for example, the ancient Roman garum (similar to a fish sauce) [21]—European cultures are less prone to adding pure MSG to their food. Nowadays, both the United States Food and Drug Administration (FDA) and the Food and Agriculture Organization of the United Nations (FAO)–World Health Organization (WHO) placed MSG in the safest category of food ingredients [22]. Nevertheless, the impression of MSG as a food additive remains [23].

In a previous study, how glutamate is perceived in some European samples was investigated [24,25]. Using a survey, a large number of healthy adult subjects in three countries representative of northern (Finland), north–central (Germany), and southern Europe (Italy) were recruited. Surprisingly, 98.4% of Italians and 98.0% of Germans could not label the taste. In Finland, the number of wrong answers decreased to 85%, which remains high compared to the Asian population [26]. Additionally, all groups primarily used the word “salt” to describe the taste of MSG. A notable taste perception and hedonic variation in the response of different countries to the taste of the umami solution was observed.

Building upon the findings of Cecchini et al. [24] and using a similar protocol but a novel artificial intelligence language processing approach in this work, all participants were asked to blind-taste MSG dissolved in water, without any other reference point, and put the perceived taste experience into words. This procedure is known as absolute taste recognition. Later in the protocol, participants compared the taste of the umami solution to that of a regular salt solution and a plain water solution in a relative taste recognition procedure. Participants were then asked to indicate the pleasantness of the solution based on a classical continuous scale and compare it to a salt solution. Demographics such as age, education level, type of residential area (big/small city or village), salt/sweets/alcohol consumption, recollection of taste preference, and smoking habits were gathered. A general descriptive analysis was conducted and a text semantic-based analysis of the sentences expressed by the entire group as well as any differences based on gender were taken into account.

Text sentiment analysis (TSA) is a text mining technique that uses machine learning and natural language processing (NLP) to analyze the emotional tone of the expressed wordings. Although the algorithm has roots in the field of computational linguistics dating back to the ’60s [27,28], it has gained significant attention and development with the rise of machine learning techniques in recent years [29]. Text semantic analysis has started to be used in various fields, including healthcare [30], sociology [31], and politics [32], to extract insights from textual data and understand emotions and moods in several disciplines, as well as in numerous applications, ranging from speech recognition [33], to information and data retrieval, to natural language processing and machine interpretation [34].

Here, TSA was used to interpret the emotional valence of the sentences used to describe the taste of the MSG solution by the participants. This approach seeks to uncover nuanced meanings that may not be captured by traditional rating scales used to measure taste intensity, pleasantness, or other characteristics, as also observed in other research fields, such as pain studies [35]. Then, a taste and perceptual profile of umami taste was established in the entire study group and within gender subgroups.

We anticipate that the representation of the umami taste will be unfamiliar and distinctive. The goal is to establish taste and perceptual umami profiles that can aid research in Western culture’s comprehension and acceptance of this exclusive taste.

## 2. Method

### 2.1. Participants

A dataset of 309 cases was collected based on a previous study [24]. From it, 76 subjects were excluded for the following reasons: 6 exhibited umami ageusia, one exhibited taste distortion in both umami and salt perception, and 69 had lived in Austria “less than two years”. All the subjects who answered the same question with “always” and “most of my life” were kept.

After this pruning, a dataset of 233 final cases was counted, on which the rest of the analysis was performed.

Participants voluntarily joined in a face-to-face interview at public locations of interest. They received no compensation but freely expressed curiosity in the citizen science survey. The survey was anonymously collected, and information unrelated to identified or identifiable natural persons was stored according to Article 4 (1) of the General Data Protection Regulation (GDPR) of the DSG (Austrian Data Protection Act). The study followed the Declaration of Helsinki and was approved by the local ethics committee of the University of Graz (GZ. 39/100/63 ex 2023/24).

### 2.2. Procedure

After receiving the information about the study, each participant was given, in a blind mode, a transparent taste solution containing monosodium glutamate (MSG, [L-glutamic acid monosodium salt monohydrate—C_5_H_8_NNaO_4_-H_2_O, Sigma-Aldrich Chemistry, Steinheim am Albuch, BW, Germany]) in water, in an absolute degree of taste recognition modality (i.e., no comparison with other tastes). They were asked to describe their perception of the umami taste in an open-answer modality, using their own words, and eventually to identify it. Subsequently, an MSG/NaCl/H_2_O discrimination test was delivered to identify MSG non-tasters (n = 6) or other taste problems (n = 1) using the protocol described in Cecchini et al. (2019) [24]. After the discrimination test, the discrimination of umami solution compared to salt and plain water solutions (relative degree of taste recognition) was performed. The relative degree of recognition approach, added to the absolute recognition approach, was chosen to control any effects due to sensory context. Indeed, If the taste recognition pattern obtained in the relative degree of recognition matched the one obtained in the absolute degree of recognition, it can be deduced that the measurement was not influenced by sensory context [36,37].

All tastants were delivered in liquid solutions with concentrations for MSG and NaCl of 241 mM and 200 mM, respectively, as determined in a previous study [38], and were found to be iso-intense and suprathreshold. The solutions were presented in 100 mL dark glass mist spray bottles. Before tasting each solution, the participants rinsed their mouths with water. The MSG/NaCl solutions were rated according to pleasantness (continuous scale of “−10” = extremely unpleasant to “+10” = extremely pleasant). Moreover, after the discrimination test, the participants were asked to label the tastes just presented (to assign a basic taste, if possible).

The following demographics were obtained: age, sex, height, weight, smoking (yes, no), type of residence (countryside, village, city, or big city), whether they had lived abroad for a time longer than two months or never, habit of adding salt to their food (“Do you generally add salt to your food at the table at the time of eating?”; four-point Likert scale, 1—never, 2—sometime, 3—most of the time, 4—always), and habit of indulging in sweet treats (“How often do you consume sweets on average?”; five-point Likert scale, 1—never, 2—once a month, 3—once a week, 4—more than once a week, 5—every day), (Table 1). Finally, we gathered information about participants’ liking of the other basic tastes (“How much do you like sweet/salty/sour/bitter/spicy food?”; five-point Likert scale, ranging from “not at all” to “very much”).

### 2.3. Statistical Analysis

Data analysis was performed using IBM SPSS statistics (IBM Corp. Released 2023. IBM SPSS Statistics for Windows, Version 29.0.2.0 Armonk, NY, USA: IBM Corp). The statistical threshold for significant results was set to *p* < 0.05. In instances where multiple comparisons were conducted, appropriate corrections to account for the increased risk of Type I errors was applied. Moreover, when the data did not meet the assumption of normality, a non-parametric statistical test or descriptors that are robust to violations of normality assumptions were used.

A text sentiment analysis (TSA), which identifies the process of affective and subjective states expressed in a text, was performed for the categorical variable describing the umami taste. The verbal descriptors of the presented taste were classified in terms of emotional expressiveness using the VADER (Valence Aware Dictionary for sEntiment Reasoning) routine trained on a gold-standard lexicon and a natural language processing (NLP) toolkit [39]. The algorithm is implemented in the Text Analytics Toolbox available in MATLAB R2022b (MathWorks, Natick, MA, USA).

## 3. Results

### 3.1. MSG Verbal Descriptors

The descriptors associated with the MSG solution were preprocessed, which included lemmatizing text and stopped words (“and”, “to”, “a”, “the”). Finally, adverbs, appositions, and punctuation were removed. A total of 547 descriptors between adjectives and nouns, with 152 unique descriptors for the entire sample, were found. Subdividing by gender resulted in a total of 282 verbal descriptors for females, 108 of which were unique, and 265 descriptors for males, 80 of which were unique. Figure 1 for all groups and Figure 2 for the two gender sub-groups depict a visual representation of the verbal descriptors in word clouds.

The most frequent word used by the entire group to describe the MSG solution was the word “sour”, with a word count = 71 (13.0%), followed by “bitter”, with a word count = 51 (9.3%), and then “salty”, with a word count = 47 (8.6%). Only a few recognized the taste of glutamate, with a word count = 8 (1.5%)—five females and three males. Other attributes with a word count bigger than/equal to 10 were “unpleasant”, with a word count = 20 (3.7%); “neutral”, with a word count = 17 (3.1%); “strange”, with a word count = 14 (2.6%); “sweet”/“tasteless”/“lemony”/“unknown”, each of which with a word count = 13 (2.4%); and “disgusting”/“mild”, both of which with a word count = 10 (1.8%). It is worth noticing that when all the words with the semantic meaning as sour were counted together, such as “sour”, “lemony”, “tart”, and “vinegar”, the total word count was 117, or 21.4% of the entire sample.

Notably, there was also a prevalence of words with a meaning associated with the persistent taste sensation of MSG. This sensation was described using terms like “aftertaste”, “intense”, and “long-lasting”.

To establish a taste sensory profile of the MSG solution in our tested group, the number of words associated with each of the five descriptors (absolute degree of taste recognition) were counted. The results were presented on a seven-dimensional radar plot, including spicy and water, alongside the five primary tastes. The findings are shown in Figure 3a.

To determine whether the taste sensory profile identified from the descriptors was consistent across different sensory contexts, we used a radar chart using MSG taste descriptors relative to salt and water collected after the discrimination tests. They are presented in Figure 3b,c. MSG taste profiles (a), (b), and (c) indicate that Austrians tried to describe the umami taste using a combination of the four basic tastes without using the word umami. On the contrary, the entire group clearly identified salt (e) and water (f) (Figure 4a,b).

The words “sour” and “salty” were consistently the most frequently mentioned tastes in both the absolute taste recognition approach and the relative taste recognition method. Surprisingly, umami was rarely mentioned, while salt and pure water were consistently identified. Additionally, it was found that different gender subgroups had similar MSG taste sensory profiles compared to the whole group.

For a more general perceptual response, the following four components, with their positive and negative connotations from the open-answer descriptions, were extracted: pleasant (unpleasant), appetizing (disgusting), familiar (unfamiliar), persistent (fleeting). Based on this categorization, it was found that the overall perception of the MSG solution was mostly unpleasant (26.6%), unfamiliar (20.0%), and disgusting (12.2%), but also persistent (8.8%) (Figure 5).

The examination of subgroups based on gender showed a shift in perceptual patterns. Females tended to perceive the solution as more unpleasant (32.5%), while males found it to be more unfamiliar (24.8%). However, the general pattern closely resembled the one observed in the entire group.

### 3.2. Text Sentiment Attribute Analysis

The TSA analysis generated numerical values ranging from −1 (extremely negative meaning of the text) to +1 (extremely positive attribute). The analysis showed that the entire group in the study described the umami taste with an overall neutral meaning (median = 0.00, IQR = −0.42) (Figure 6a), with no statistical difference between the sex subgroups (*p* = 0.807, males (median) = 0.00, IQR = −0.36, females (median) = 0.00, IQR = −0.42).

Then, the numerical values were categorized into positive (>0), neutral (= 0), and negative (<0) sentiment attributes. According to our findings, the overall sample had 38% expressing a negative, 45% a neutral, and 17% a positive sentiment toward savoring the MSG solution (Figure 6b—all).

No statistically significant difference was noticed by subdividing the entire sample into males and females (Figure 6b—f,m).

TSA analysis generated numerical values that were uncorrelated with education, residence, or experience abroad.

### 3.3. Pleasantness Ratings

Participants rated the umami solution, M (median) = −1.00, as less pleasant than the salt solution, M = 0.00 (Z_(WilcoxonSumRank)_ = −2.938, *p* = 0.003). A non-parametric test of differences among measures of pleasantness was conducted for the subgroup individuated by gender. It was observed that only the male group rated the MSG (M = 0.00) as less pleasant than the NaCl (M = 0.05) (Z(WSR) = −2.545, *p* < 0.011), while the female group did not (Z(WSR) = −1.578, *p* < 0.115). In comparison, males and females rated the NaCl and MSG as having similar pleasantness. Data are visualized in the Appendix A.

Pleasantness ratings for salt and MSG were positively correlated (r_Spearman_ = 0.316, 95% confidence interval (CI) = 0.192 ÷ 0.431, *p* < 0.001). Dividing the groups by gender, the correlation between variables remained significant in the male (rs = 0.419, CI = 0.251 ÷ 0.536, *p* < 0.001) as well as in the female (rs = 0.197, CI = 0.011 ÷ 0.370, *p* = 0.033) subgroup.

Whether the monotonic association identified by the Spearman rho correlation between the NaCl pleasantness ratings and the MSG pleasantness ratings was specifically linear was investigated. Linearity was established by visual inspection of both variables on a scatterplot. There was homoscedasticity and normality of the residuals. Then, with a regression model, how the pleasantness of one taste predicted the other was estimated. It was found that the NaCl pleasantness rating statistically significantly predicted MSG pleasantness ratings, F (1, 231) = 24.34, *p* < 0.001, accounting for 9.5% of the variation in MSG pleasantness with adjusted R^2^ = 9.1%, a medium size effect according to Cohen (1988) [40] (Figure 7).

Pleasantness ratings were uncorrelated with education, residence, experience abroad, or age.

### 3.4. Recalled Taste Preference

The participants answered the questions, “How much do you like to eat sweet/salty/sour/bitter/spicy food?” as follows (median, IQR): sweet (4.00, 2.00), salty (4.00, 1.50), sour (3.00, 2.00), bitter (2.00, 1.00), and spicy (4.00, 2.00). A Friedman test was conducted to determine whether there were any differences in taste preference scores for sweet, salt, bitter, sour, and spicy. The results revealed a significant difference, with test results of χ^2^(4) = 253.916, *p* < 0.001. Post hoc comparison, using Bonferroni correction for multiple comparisons, indicated that the entire group significantly preferred sweet (*p* < 0.001), salty (*p* < 0.001), sour (*p* < 0.001), and spicy (*p* < 0.001) over bitter; sweet (*p* = 0.001) and salty (*p* < 0.001) over sour; and sweet (*p* = 0.004) and salty (*p* = 0.001) over spicy (Figure 8).

After splitting the data by gender, the following group taste preference values were obtained [Median (IQR)] for females: sweet = 4.00 (1.50), salty = 4.00 (2.00), sour = 3.00 (2.00), bitter = 2.00 (0.50), and spicy = 3.00 (2.00), and for males: sweet = 4.00 (1.00), salty = 4.00 (1.00), sour = 3.00 (2.00), bitter = 2.00 (1.00), and spicy = 4.00 (1.00). A Friedman test was conducted to analyze the taste preferences of males and females separately. The male sample showed a statistically significant difference in taste preference (χ^2^(4) = 117.760, *p* < 0.001). A post hoc comparison, with Bonferroni correction, indicated that the male group significantly preferred sweet (*p* < 0.001), salty (*p* < 0.001), sour (*p* = 0.005), and spicy (*p* < 0.001) to bitter, sweet (*p* = 0.007), and salty (*p* < 0.001), and preferred spicy (*p* = 0.001) to sour. For females, the Friedman test was also significant (χ^2^(4) = 159.124, *p* < 0.001). Post hoc comparison, with Bonferroni correction, revealed that the female group preferred sweet (*p* < 0.001), salty (*p* < 0.001), sour (*p* < 0.001), and spicy (*p* < 0.001) to bitter; sweet (*p* < 0.001) and salty (*p* < 0.001) to sour; and finally, sweet (*p* < 0.001) and salty (*p* < 0.001) to spicy. A series of Mann–Whitney U tests were conducted to compare the gender differences in taste preference. The results showed that females preferred sweet (*p* < 0.001) and relatively salty (*p* = 0.025) compared to males. On the other hand, males preferred spicy (*p* < 0.001) more than females did. Box plots and statistics are reported in Figure 8.

## 4. Discussion

Our study reveals important insights regarding the sensory perception of umami taste in Austria. As expected, the group studied was unfamiliar with the umami taste and rarely could recall it by name. Moreover, a stable pattern of umami taste profile emerged, and this pattern remained consistent across different sensory contexts. Other crucial aspects of the results of this study are discussed in detail in light of existing evidence in the following paragraphs.

### 4.1. Perception of Umami Solution Through Verbal Descriptors

Part of the study was to identify verbal descriptors to express the perception of the umami solution. While numerical scales provide a limited understanding of the nuanced nature of hedonic valence, verbal descriptors can convey more subtle meanings and dimensions of a sensation. Regarding the specific MSG taste, the results highlighted a wide range of responses, with a certain degree of complexity.

A perceptual and taste profile of MSG was delineated that visually represents the overall perception of MSG and highlights its characteristics in a multidimensional format (Figure 3 and Figure 5).

#### 4.1.1. MSG Taste Profile

Our group emerged with a defined MSG taste profile (Figure 3a–c), indicating that individuals, despite their general unfamiliarity, could relate the taste of MSG to known tastes. The profile also remained consistent across different sensory contexts, confirming its reliability in the related perception.

At the top of the sensory descriptors, people used “sour”, “bitter”, and “salty” (Figure 1 and Figure 3). However, descriptors such as “sweet”, “watery”, or “spicy” were also found. At the same time, the word “umami” is not mentioned in 98.5% of the cases. Indeed, only 8 people out of 233 used the word “umami/glutamate”.

As a matter of fact, umami is not a new experience in Austrian culinary culture. Foods such as Tyrolean bacon, cured ham, and dry-cured pork loin have umami content that can reach 340 mg/100 g depending on the ripening process, and vegetable extracts (broth), stock (in cubes or powder), bouillon [41], mushroom (minimum of 40 up to 273.6 mg of umami per 100 g), and cheese [42,43] are also all essential elements of classic Austrian cuisine [44,45]. Therefore, Austrians may have an unrecognized semantic issue due to the lack of knowledge about the concept of umami as a primary taste. This finding is consistent with our previous research in other European countries [24,46].

The most common taste descriptor for MSG was “sour”. This contrasts with the most used taste descriptor by people interviewed in Finland, Germany, and Italy: “salty” [24]. However, when considering the second and third descriptors, the Austrian group was more aligned with the German group, using “bitter”, “salty”, and “sour”, but quite different from the Finnish group, which used terms such as “salty”, “umami” and “meat”/“meaty”.

The association of “sour” to describe the MSG solution may be linked to the increased salivation elicited by MSG, which is similar to sour substances. It has been found that salivation induced by MSG ranked second only to that induced by sour taste [47]; this similarity may contribute to our subjects’ re-evocation of the sour taste experience. Interestingly, this effect can be crucial in the application of treatment for dry mouth, and some studies with positive results are already available [48] (Satoh-Kuriwada & Sasano, 2015)

The recalling of sweet and salty tastes can be attributed, at least in part, to the sodium component of MSG, because a low concentration of NaCl can be confused as sweet when subjects are adapted to water or a neutral taste like saliva [49].

Finally, in the MSG taste profile, the taste “water” was also included, in which all terms such as “tasteless”, “bland”, “watery”, and “water” were grouped. It is important here to stress that the possibility that the subjects who described the taste as “tasteless” had umami ageusia was ruled out by the differentiation test included in the protocol between the MSG solution and water, as well as between the MSG and NaCl solutions. However, behavioral and psychophysical studies have shown variability in human perception of the umami taste. Aside from umami ageusia, which indicates a disorder that makes individuals unable to taste umami, Lugaz, Pillias, and Faurion [50] (2002) also classified a category of people indicated as “hypo-tasters” who could perceive MSG at relatively high concentrations. They estimated that around 10% of the general population are hypo-tasters. In our study, approximately 9.0% (21 subjects) described the MSG solution as a neutral/bland taste, confirming Lugaz and colleagues’ findings [50]. The variability in human umami taste perception still needs to be fully understood. Genetic mechanisms may play a role in these variations [51,52,53], but factors such as age, body weight [54], or previous exposure and familiarity with umami taste [55,56,57] may also be influential.

#### 4.1.2. MSG Perceptual Profile

The perceptual profile of MSG was mainly characterized by “unpleasant” and “unfamiliar” components (Figure 5).

The perception of unpleasantness is understandable because tastes are rarely experienced in isolation. An authentic eating experience involves tasting, smelling, seeing, and feeling the texture and temperature of the food [58,59]. Additionally, unlike other taste qualities, relatively pure MSG solutions are not commonly found in nature [60].

In the MSG perceptual profile, terms such as “unfamiliar”, “unknown”, “difficult to tell”, “strange”, or “not sure” were grouped under the “unfamiliar” category. Here, the lack of experience with the definition of this taste among the entire group has resulted in a reduced association with the foods that contain umami. In addition to the unfamiliarity with the umami taste, the complexity of this taste may also play a role. For instance, it is known that when combined with foods, umami can blend in a way that creates a masking effect [6].

#### 4.1.3. Group Sentiment Toward the MSG Solution

Using text sentiment analysis (TSA), participants’ subjective verbal feedback on the MSG solution was transformed into numerical values. TSA involves analyzing verbal responses to quantify emotional undertones, assigning to the verbal responses a continuous index from +1, extremely positive, to −1, extremely negative. The analysis revealed that the general sentiment towards the MSG solution was mainly neutral, with a wide variance of sentiments from negative to positive (Figure 6a). The overall neutral sentiments towards the MSG solution were also maintained in the male and female subgroups (Figure 6b), which were mostly neutral, with percentages of 49% and 41%, respectively. This result is a crucial finding because the neutral sentiment suggests a potential for acceptance, given the corrected information is provided to overcome the unfamiliarity barrier. On the other side, the negative sentiments, at ca. 38%, present also in the sex subgroups, imply that there was a solid negative acceptance toward MSG among the participants, which should be addressed with the correct information. However, considering that the positive sentiments were also relevantly significant and that the neutral was the majority, with appropriate exposure and education, individuals may grow to appreciate the taste of MSG. Finally, there were no observed significant gender differences in the acceptance and evaluation of umami across gender lines.

### 4.2. MSG and NaCl Direct Comparison

Our study found a significant positive linear relationship between the pleasantness of MSG and the salt solution. This implies that the perceived pleasantness of MSG can be predicted based on the more familiar taste of salt.

As a matter of fact, when evaluated independently, salty and umami tastes exhibit several similar properties. They both contain a sodium component, their temporal time-intensity profiles and temporal dominance of sensation are similar, and the duration of the taste solution increases in a concentration-dependent manner [61]. The interaction between saltiness and umami has been a subject of examination over the years, and food processing industries have largely used the enhanced palatability from MSG depending on NaCl and vice versa [7,11,62,63,64]. The shared properties of the two tastes may be the driver of the linear and positive relationship found in our data.

This finding bears significant practical implications for manipulating sodium content in food. The saltiness of MSG is approximately 30% that of NaCl by molar sodium concentration and 10% that of NaCl by weight [6,65]. By leveraging the synergic actions of the combination of the two, it is possible to obtain a significant salt reduction without compromising palatability [6,62,66,67,68].

There were no significant gender differences in the linear relationship between MSG and NaCl pleasantness; both retained a significant linear relationship, indicating that the conclusions drawn are applicable across gender lines.

As people age, the intensity ratings for all five taste qualities decrease, depending on the taste quality; it is most pronounced for bitter and sour tastes, while it is least pronounced for umami [69] (Barragán et al., 2018). This finding may also explain the observed independence between pleasantness and age.

### 4.3. Taste Preference Expressed as Recalled Experience

When preferences were expressed based on real-life experiences, salt was not found to be unpleasant, but was rated as one of the most preferred, similar to sugar (Figure 8). This confirms that taste preference goes beyond the extent of tastants and is influenced by a range of variables, among which are cultural background [70], personal experience [71], genetic factors [72], smell [73], texture [74], and sensory context [75,76,77]. Since salt, as with all tastants, is rarely consumed alone, its isolated unpleasantness measured in a lab environment does not reflect its real-world preference, suggesting that this can also be the case for umami.

Interestingly, a clear sex-dependent taste preference in the recalled taste preference was observed. Females tended to favor sweeter and saltier food, while males leaned towards spicy food. This is consistent with existing research, which has found that women often consume more sweet or high-calorie meals than men [78] and those sweet products are often marketed toward them [79]. The influence of sex hormones on taste inclination due to a different body composition from males is a plausible explanation [80,81]. Our results also support studies that have observed that males prefer spicy food more than females do [82], and in some cultural contexts, the consumption of chili peppers has been related to strength, daring, and masculine personality traits [83], as well as physiologically, as shown in a study linking salivary testosterone and the quantity of hot sauce individuals consumed in a lab meal [84].

### 4.4. Umami Ageusia in the Austrian Group

In our original sample of 309 subjects, 1.9% (6 subjects) were unable to taste umami. This percentage aligns with the findings of Cecchini (2019) [24] when considering the entire group in the study. However, there was a variation in the prevalence of umami non-tasters in the three countries in the study. Specifically, Finland, Germany, and Italy had percentages of non-tasters of 1.3%, 3.3%, and 0.4% respectively. In the Norwegian population, the percentage of non-tasters was 4.6% [46], while the first study identifying this taste disorder found a 3.5% prevalence of non-tasters in the French population [50].

These results indicate that variations in umami perception are more likely due to individual differences rather than differences in the population as a whole. However, the association between umami taste phenotypes and variations in umami taste receptor genes remains unclear at this point, and further studies are needed [85].

### 4.5. Research Challenge and Future Directions

The participants in this study were from an urban area in central Austria. Therefore, other groups, such as those living in more rural areas, may have different psychological and physiological drivers of dietary patterns. This underscores the need for further research into umami taste perception within various environmental segments. It is also important to acknowledge that our findings should be considered in light of the exploratory nature of the survey.

Future prospective of umami sensory perception in Europe would be to plan a strategy to familiarize Western countries with the umami taste. Additionally, given the findings that umami’s most commonly associated descriptor is sour, followed by salty and bitter, it would be worthwhile to explore the relationship between the sensory perception of umami—particularly, its pleasantness—and the other taste profiles, as we have done for salt, namely, sour, bitter, and sweet.

### 4.6. Conclusions

In conclusion, our study group has identified a clear need for increased awareness of umami. By growing exposure and understanding of MSG, it should be possible to shift perception towards a more positive reception, mirroring the widespread acceptance of salt.

## Figures and Tables

**Figure 1 foods-14-00022-f001:**
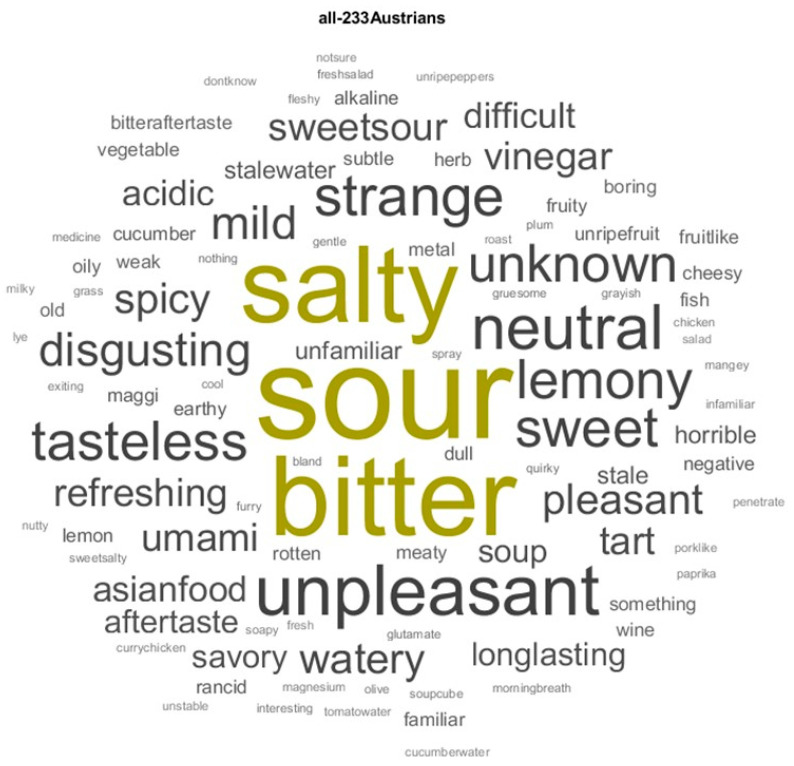
Word cloud based on the term-frequency counter of the text mining analysis for the entire group. The more a specific word appears in a source of textual data, the bigger and bolder it appears in the word cloud. The descriptors with a frequency of appearance of more than 40 are portrayed in olive.

**Figure 2 foods-14-00022-f002:**
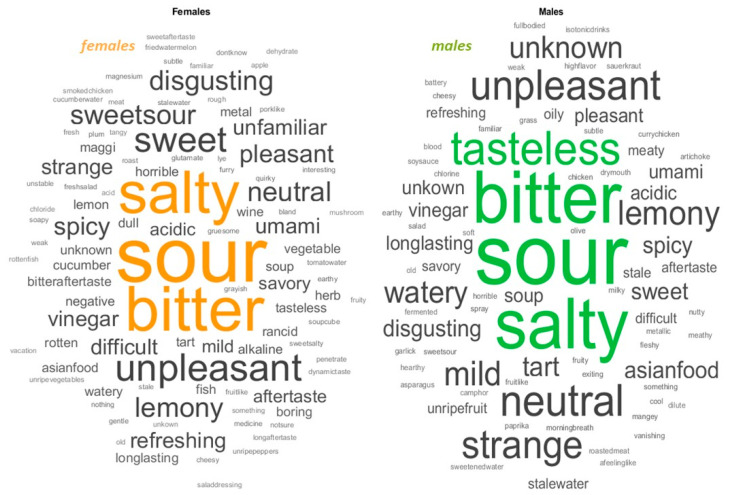
Word cloud based on the term-frequency counter of the text mining analysis separately for the male (orange) and female (green) subgroups. The more often a specific word appears in a source of textual data, the bigger and bolder it appears in the word cloud. The highlighted words have a minimum frequency of appearance of 10.

**Figure 3 foods-14-00022-f003:**
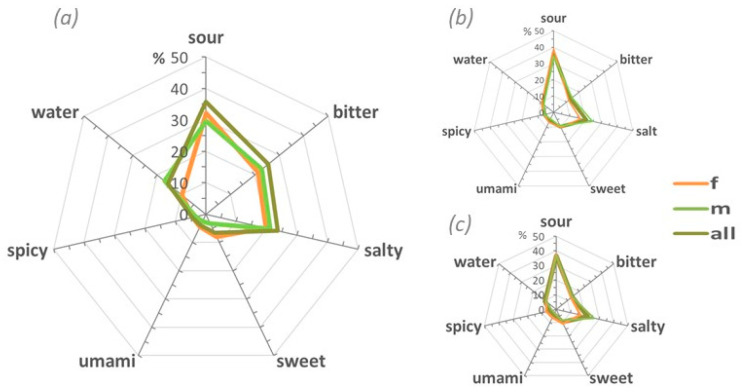
Radar charts of MSG taste profiles. Radar charts representing the taste perception described by the samples in the study. The umami taste description, whether extracted by the open-answer descriptors (**a**) or obtained in a test in comparison to a salt (**b**) or simple water solution (**c**), shows a consistent pattern across the whole group (all), subgroups (m, f), and tests (**b,c**). This consistency indicates that the umami taste was described as a complex combination of primary tastes with a predominancy of sour, bitter, and salty flavors, while umami itself remained unidentified. The scale, in percentage, refers to the total number of subjects in the group.

**Figure 4 foods-14-00022-f004:**
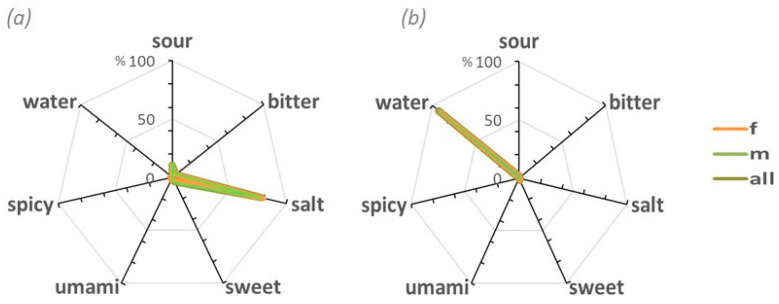
Radar charts of (**a**) NaCl solution and (**b**) tasteless solution taste profiles. The sample clearly identified the respective tastes for salt (90%) and water (97%), with no differences between females and males. The scale, in percentage, refers to the total number of subjects in the group.

**Figure 5 foods-14-00022-f005:**
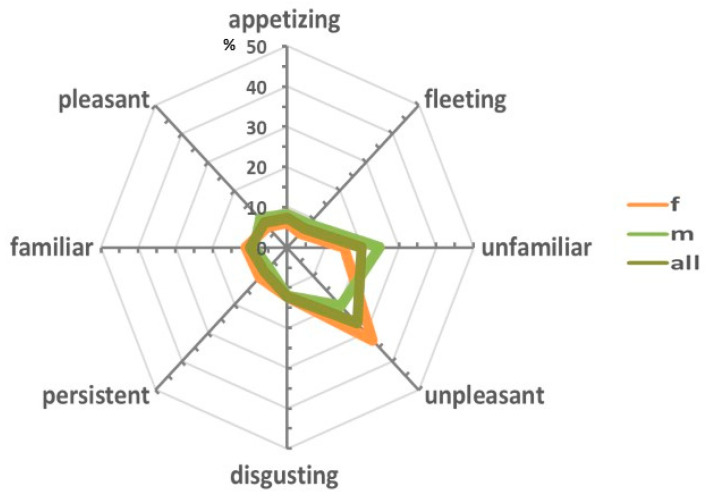
Radar charts of MSG hedonic perception described by the samples in the study. The umami perceptual profile is reported in four dimensions—appetizing (disgusting), pleasant (unpleasant), familiar (unfamiliar), and persistent (fleeting)—with opposite valence on the extremities of the axis. The categories were based on the open-answer descriptors. Overall, the entire group described the MSG taste as unpleasant and unfamiliar. In the charts, the entire group is depicted in olive green (all), while the female and male subgroups are shown in orange (f) and green (m), respectively. In the chart, the entire population is depicted in olive green, while the female and male subgroups are shown in orange (f) and green (m), respectively. The scale, in percentage, refers to the total number of subjects in the group.

**Figure 6 foods-14-00022-f006:**
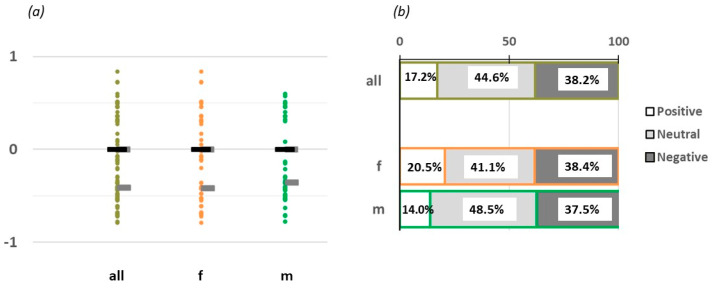
Text sentiment analysis (TSA) results. (**a**) TSA of the phrases pronounced in the open-answer description of the MSG solution. The median (black bar) and IRQ (grey bars) report on the point distribution. (**b**) TSA percentage of the hedonic components: positive (white), neutral (grey), and negative (black). All: entire group, f: females, m: male.

**Figure 7 foods-14-00022-f007:**
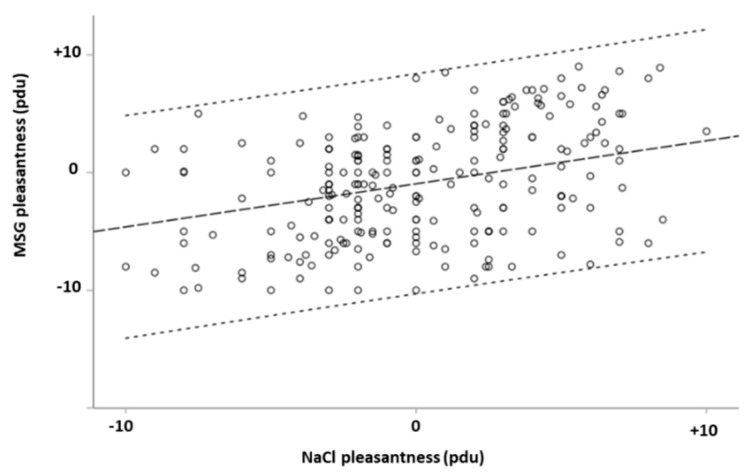
Linear regression analysis of MSG pleasantness as a function of NaCl pleasantness. The dashed line represents the best-fit linear regression model. The area around the regression line delineated by the dotted lines indicates the 95% confidence interval of the predicted MSG pleasantness values based on NaCl pleasantness ratings. The model explains 9.5% of the variance in MSG pleasantness (R^2^ = 0.095), indicating a statistically significant linear relationship (*p* < 0.001) with coefficient β = 0.37. pdu: procedure defined unit.

**Figure 8 foods-14-00022-f008:**
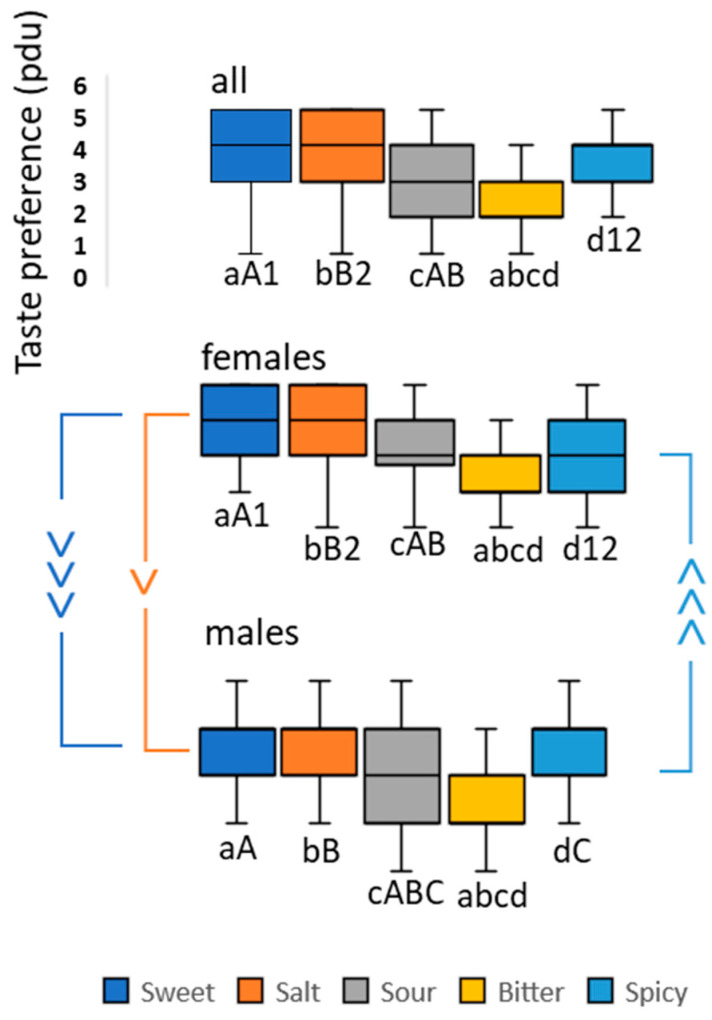
Box plots (median, IQR) of taste preference expressed as recalled experience on the Likert scale (0–6) for sweet, salty, sour, bitter, and spicy, organized in rows by group (all: entire group), and gender, and in columns by taste (sweet, salty, sour, bitter, and spicy). The statistical significance levels between groups are highlighted with “>“, where the angle bracket’s opening is in the greater value’s direction. Also: “>>>“, and “>” indicate *p* < 0.001, *p* < 0.01, and *p* < 0.05, respectively. Statistical differences within groups are highlighted by letters/numbers; the same letter/number indicates statistically significant differences.

**Table 1 foods-14-00022-t001:** Participants’ characteristics.

Demographics					
	All	f	m	*p*-Value	Chi-Square Test (df)
Sex ratio	233	117	116	n.a	n.a
Age (mean years)	27.4 (SD = 10.5)	26.5 (SD = 10.7)	28.3 (SD = 10.3)	0.203	−1.275 (231) ^$^
Smoking	Non-smoker: 186 (80%)	100 (85%)	86 (74%)	0.02 *	7.561 (2)
	Smoker: 47 (20%)	17 (15%)	30 (26%)		
Education	At least high-school diploma 145 (62%)	71 (61%)	74 (64%)	0.88	1.222 (4)
Residence	In a big city (pop. > 100,000): 160 (60%)	72 (62%)	88 (76%)	0.06	5.731 (2)
	In a city (pop. = 5000 ÷ 100,000): 44 (19%)	28 (24%)	16 (14%)		
	In a village/rural area (pop. < 5000) 29 (12.4%)	17 (15%)	12 (10%)		
Having lived abroad	Experience of living in a foreign country 56 (24%)	35 (30%)	21 (18%)	0.04 *	4.451 (1)
	Never lived abroad 117 (76%)	82 (70%)	95 (82%)		
Salt consumption	Never: 58 (25%)	29 (25%)	29 (25%)	0.993	0.087 (3)
	Sometimes: 108 (46%)	54 (46%)	54 (47%)		
	Most of the time: 56 (24%)	28 (24%)	28 (24%)		
	Always: 11 (5%)	6 (5%)	5 (4%)		
Sweets consumption	Never: 11 (4.7%)	3 (3%)	8 (7%)	0.514	3.269 (4)
	Once a month: 8 (3.4%)	3 (3%)	5 (4%)		
	Once a week: 69 (30%)	35 (30%)	34 (29%)		
	More than once a week: 105 (45%)	54 (46%)	51 (44%)		
	Every day: 40 (17%)	22 (19%)	18 (16%)		

Footnote: SD: standard deviation, n.a: not applicable, df: degree of freedom, * indicates significant differences, *p* < 0.05, ^$^ independent sample *t*-test.

## Data Availability

The original contributions presented in the study are included in the article/Appendix A, further inquiries can be directed to the corresponding author.

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
