# Peer review of "A Survey on the Evaluation of Monosodium Glutamate (MSG) Taste in Austria"

_foods, 2024, doi:10.3390/foods14010022_

Round 1
Reviewer 1 Report
Comments and Suggestions for Authors
This paper studies the sensory evaluation of different MSG, which is interesting, but there are some problems that can be modified.
1. The abstract is not recommended to be written in segments, and does not describe the background, significance, and means of the research.
2. The reference literature is too old, and the literature in the past three years is very few.
3. Are the 233 testers selected by the author different in age? Whether age has an effect on the test results, I think it has something to do with the light or rich taste.
4. The conclusion is too long and is more like a discussion than a conclusion. It is recommended to describe it separately.
Author Response
We appreciate the reviewers' valuable comments. In response to their suggestions, we have modified the abstract, improved the introduction by incorporating relevant and recent references, refining the methods and discussion section, and removing material from the conclusion that is more appropriate for the discussion section. Additionally, we have addressed each point raised by the reviewer in detail in the text below. For your convenience, the manuscript file with tracked changes (extension: _tracks) and the file with the integrated changes without tracks (extension: _clear) are attached.
Review 1
This paper studies the sensory evaluation of different MSG, which is interesting, but there are some problems that can be modified.
- The abstract is not recommended to be written in segments, and does not describe the background, significance, and means of the research. --> response to point 1: In response to the reviewer's suggestion, we have revised the abstract to make it less segmented while still adhering to the structured abstract format (Background, Methods, Results, and Conclusion without headings) recommended in the journal guidelines. Additionally, we have emphasized the background information and the significance of the work. For your convenience we report the new abstract here below: “Abstract: The umami taste is well-validated in Asian culture but remains less recognized and accepted in European cultures despite its presence in natural local products. This study explored the sensory and emotional perceptions of umami in 233 Austrian participants who had lived in Austria for most of their lives. Using blind tasting, participants evaluated monosodium glutamate (MSG) dissolved in water, providing open-ended verbal descriptions, pleasantness ratings, and comparisons to a sodium chloride (NaCl) solution. Discrimination tests excluded MSG-ageusia, and basic demographic data were collected. A text-semantic-based analysis (TSA) was employed to analyze the emotional valence and descriptive content of participants' responses. The results showed that MSG was predominantly associated with neutral sentiments across the group, including both female and male subgroups. "Sour" was the most frequent taste descriptor, while "unfamiliar" characterized the perceptual experience. Pleasantness ratings for MSG and NaCl were positively correlated, suggesting that overcoming the unfamiliarity of umami could enhance its acceptance and align it with the pleasantness of salt. These findings advance the understanding of umami sensory perception and its emotional and cultural acceptance in European contexts, offering valuable insights for integrating umami into Western dietary and sensory frameworks.”
- The reference literature is too old, and the literature in the past three years is very few. →respons to point 2-->We thank the reviewer for the suggestion. We have accordingly upgraded the bibliography with more recent scientific papers, which we list below for their reference: â–ª Ding, Y., Liu, Y., & Xu, S. (2024). How brand gender affects consumer preference for sweet food: The role of the association between gender and taste. Psychology & Marketing â–ª Elman, I., Ariely, D., Tsoy-Podosenin, M., Verbitskaya, E., Wahlgren, V., Wang, A. L., ... & Krupitsky, E. (2023). Contextual processing and its alterations in patients with addictive disorders. Addiction Neuroscience, 7, 100100.
â–ª Garaus, M., Weismayer, C., & Steiner, E. (2023). Is texture the new taste? The effect of sensory food descriptors on restaurant menus on visit intentions. British Food Journal, 125(10), 3817-3831. â–ª Lee, B. P., & Spence, C. (2023). Synergistic combination of visual features in vision–taste crossmodal correspondences. Multisensory Research, 36(7), 573-612. â–ª Li, J., Zhong, F., Spence, C., & Xia, Y. (2024). Synergistic effect of combining umami substances enhances perceived saltiness. Food Research International, 189, 114516. â–ª Ma, F., Li, Y., Zhang, Y., Zhang, Q., Li, X., Cao, Q., ... & Liu, G. (2024). Effects of umami substances as taste enhancers on salt reduction in meat products: A review. Food Research International, 114248. â–ª Morita, R., Ohta, M., Umeki, Y., Nanri, A., Tsuchihashi, T., & Hayabuchi, H. (2021). Effect of monosodium glutamate on saltiness and palatability ratings of low-salt solutions in Japanese adults according to their early salt exposure or salty taste prefer-ence. Nutrients, 13(2), 577. â–ª Motoki, K., Spence, C., & Velasco, C. (2023). When visual cues influence taste/flavour perception: A systematic review and the critical appraisal of multisensory flavour perception. Food Quality and Preference, 104996. â–ª Muscogiuri, G., Verde, L., Vetrani, C., Barrea, L., Savastano, S., & Colao, A. (2024). Obesity: a gender-view. Journal of Endo-crinological Investigation, 47(2), 299-306. â–ª Walker, J. C., & Dando, R. (2023). Sodium Replacement with KCl and MSG: Attitudes, Perception and Acceptance in Reduced Salt Soups. Foods, 12(10), 2063. â–ª Wu, B., Eldeghaidy, S., Ayed, C., Fisk, I. D., Hewson, L., & Liu, Y. (2022). Mechanisms of umami taste perception: From molecular level to brain imaging. Critical Reviews in Food Science and Nutrition, 62(25), 7015–7024. â–ª Yoshida, R., & Ninomiya, Y. (2023). Umami and MSG. In Umami: Taste for Health (pp. 7-42). Cham: Springer International Publishing â–ª Zhu, Y., Liu, J., & Liu, Y. (2023). Understanding the relationship between umami taste sensitivity and genetics, food-related behavior, and nutrition. Current Opinion in Food Science, 50, 100980. - Are the 233 testers selected by the author different in age? Whether age has an effect on the test results, I think it has something to do with the light or rich taste. →response to point 3--> The volunteers' ages ranged from 17 to 64 years old, and we did not find any correlation between age and pleasantness. Assessing intensity or threshold sensitivity was beyond the scope of our study, and we chose a suprathreshold concentration to be easily detected. However, as suggested by the reviewer the loss of sensitivity with age could be an important confounding factor. To clarify this, we added the following paragraph (lines 475 to 479): "Barragán and his colleagues found that as people age, the intensity ratings for all five taste qualities decrease. They observed that this decrease varies by taste quality; it is most pronounced for bitter and sour tastes, while it is least pronounced for umami (Barragán et al., 2018). This finding may also explain the observed independence between pleasantness and age.".
- The conclusion is too long and is more like a discussion than a conclusion. It is recommended to describe it separately.
→response to point 4 --> We thank the reviewer for their suggestion. We have revised the conclusion accordingly, and now the paragraph reads (lines 530-533):“ In conclusion, our study group has identified a clear need for increased awareness of umami, an essential component of taste that can contribute to healthier eating habits. By growing exposure and understanding of MSG, it should be possible to shift perception towards a more positive reception, mirroring the widespread acceptance of salt.”
Please see the attachment.

Reviewer 2 Report
Comments and Suggestions for Authors
This manuscript was entitled as “A Survey on the Evaluation of Monosodium Glutamate (MSG) Taste in Austria.” This study aimed to explore the perception of unami taste in a Austrian population. The authors concluded that their study extensively contributed to understanding umami sensory perception and its acceptance in a European group.
There are some major concerns about this manuscript.
.
1. When the authors compared the unami taste with salty taste in their study, they found the sensory perception of unami is similar to that salt. If the authors compare the unami with other tastes such as sour taste, whether the results are different?
2. Since this study was performed in local Austrian people, I worry about whether the results can be generalized to other population.
Reviewer 3 Report
Comments and Suggestions for Authors
Please find a detailed comment file attached

The authors frequently use words such as "we" and "our," which are not appropriate in a scientific presentation. These should be replaced with neutral phrasing throughout the text, avoiding the use of such pronouns. Additionally, providing recommendations is challenging without line numbering.
Round 2
Reviewer 1 Report
Comments and Suggestions for Authors
The author has made a lot of modifications based on the reviewer's comments, which is great, but there is still a small problem that the author needs to modify. The lines in Figure 3, Figure 4 and Figure 5 are too thick, which affects reading, and the color matching is not beautiful enough.
Reviewer 2 Report
Comments and Suggestions for Authors
Thanks for your responses. I do not have further comment.
Reviewer 3 Report
Comments and Suggestions for Authors
The manuscript has generally improved in quality after the first round of review. It is now more consistent and easier to read. However, there are still formatting issues that require careful attention. I recommend a thorough review involving at least three readers to ensure accuracy.
Specific Remarks:
-
Table 1:
- The total number provided is 233, but the gender split (117 + 118) sums to 235, indicating an error.
- Additionally, the other numbers presented add up to 116 males. Please correct these inconsistencies.
-
Figure and Text References:
- There are still mismatches between figure numbering and in-text references. For example, "Figure 7" is mentioned on line 353, but no corresponding Figure 7 is present.
- Please carefully verify all in-text references to tables and figures to ensure consistency and accuracy.
-
Figure 5:
- Avoid starting the accompanying text in bold format. Ensure consistent styling throughout the manuscript.
-
References:
- There are discrepancies between in-text citations and the reference list. For example:
- "Han et al., 2018" on line 467 and "Shigemura et al., 2009" on line 465 are either incorrectly formatted or missing from the reference list.
- Ensure all references are accurately cited in the text and correctly included in the reference list.
- On line 527, "Barragán and his colleagues" is mentioned for the first time. Cite this reference directly and correctly at this point.
- There are discrepancies between in-text citations and the reference list. For example:
Overall Comment:
These formatting and referencing issues must be addressed thoroughly to maintain the manuscript's value and credibility for readers.
